# A cross-sectional analysis of clinicopathologic similarities and differences between Henoch-Schönlein purpura nephritis and IgA nephropathy

**Motonori Sugiyama[☯], Yukihiro Wada[ID]\*[☯], Nobuhiro Kanazawa, Shohei Tachibana, Taihei Suzuki, Kei Matsumoto, Masayuki Iyoda, Hirokazu Honda, Takanori Shibata**

Division of Nephrology, Department of Medicine, Showa University School of Medicine, Tokyo, Japan

[☯] These authors contributed equally to this work.
\* yukihiro@med.showa-u.ac.jp

**Data Availability Statement:** All relevant data are within the manuscript and its Supporting Information files.

## Abstract

### Introduction

Recent studies noted that Henoch-Schönlein purpura nephritis (HSPN) and IgA nephropathy (IgAN) share the feature of galactose-deficient IgA1 (Gd-IgA1)-oriented pathogenesis, although there are distinct clinical differences. We aimed to clarify the clinicopathologic differences between these 2 diseases.

### Methods

We cross-sectionally analyzed adult patients with HSPN (n = 24) or IgAN (n = 56) who underwent renal biopsy (RB) between 2008 and 2018 at Showa University Hospital. Serum Gd-IgA1 (s-Gd-IgA1) levels at the time of RB were compared among study groups using enzyme-linked immunosorbent assay (ELISA) with anti-human Gd-IgA1-specific monoclonal antibody (KM55). We also immunohistochemically stained paraffin-embedded sections for glomerular Gd-IgA1 (g-Gd-IgA1)-deposition using KM55. Serum inflammatory cytokines were measured using ELISA.

### Results

Glomerular endothelial injury with subendothelial IgA deposition was significant in patients with HSPN. Serum IL-8, MCP-1, TNF-α, and IL-6 levels were significantly higher in patients with HSPN than IgAN. Levels of s-Gd-IgA1 were comparable among patients with HSPN and IgAN, and a similar degree of g-Gd-IgA1-deposition was detected in both diseases. Furthermore, g-Gd-IgA1-deposition was evident in patients with histopathologically advanced HSPN or IgAN. In HSPN, significant positive correlations between s-Gd-IgA1 levels and crescent formation or IL-6 elevation were confirmed, and g-Gd-IgA1 intensity showed a significant positive correlation with MCP-1 and a tendency to positively correlate with IL-8. Meanwhile, patients with IgAN showed no correlation between inflammatory cytokines and both-Gd-IgA1. Moreover, most g-Gd-IgA1-positive areas were not double stained with CD31 in HSPN.

**Funding:** The authors received no funding for this work.

**Competing interests:** The authors have no conflicts of interest to declare.

## Conclusions

Although assessing both-Gd-IgA1 alone was insufficient to distinguish between HSPN and IgAN, patients with HSPN showed considerable glomerular capillaritis with subendothelial IgA deposition and significant elevation of serum inflammatory cytokines. Furthermore, such glomerular subendothelial IgA deposition might not contain Gd-IgA1, and factors associated with Gd-IgA1 were inconsistent among these 2 diseases. Thus, developmental mechanisms for IgAN might not apply to HSPN completely, and these 2 diseases still have different aspects.

## Introduction

Henoch-Schönlein purpura (HSP), recently also called immunoglobulin A vasculitis (IgAV), is a systemic vasculitis characterized by the deposition of IgA immune complexes (IC) in small vessels of the skin and other organs such as the gastrointestinal tract, joints, and kidney [1, 2]. Renal involvement in IgAV, also referred to as Henoch-Schonlein purpura nephritis (HSPN), is the most common and severe complication and a major factor affecting the long-term outcome of patients with HSP [3]. The annual incidence of HSP in children is estimated to be 3–26.7 cases per 100,000 for children and infants, whereas that in adults is only 0.8–1.8 per 100,000 for adults [4–6]. However, HSPN occurs more frequently in adults than children [3, 7, 8]. In addition, the clinical presentation and renal outcome can be more severe in adults than children [9]. Furthermore, the risk for progression to chronic renal insufficiency is much higher in adults at approximately 30% [3, 9, 10], and approximately 10% of adults with HSPN reach end-stage kidney disease (ESKD) within 15 years [3]. Therefore, adult-onset HSPN is characterized by formidable glomerulonephritis (GN) as well as IgA nephropathy (IgAN).

IgAN is the most prevalent type of GN worldwide [11]. Progressive glomerular and interstitial sclerosis in severe IgAN leads to ESKD in 30–40% of patients within 20 years after diagnosis [12, 13]. Histopathologically, IgAN is characterized by mesangial cell proliferation with IgA-IC deposition in the glomerular mesangium, which is indistinguishable from pathologic findings of HSPN [2, 12, 14]. Regarding the pathogenesis of IgAN, several studies that investigated aberrant IgA1 O-glycosylation indicated that galactose-deficient IgA1 (Gd-IgA1) plays a pivotal role in the progression of IgAN [15–23]. According to these studies, patients with IgAN have aberrant IgA1 molecules with a Gal deficiency of O-linked glycans in the hinge region, which indicates that Gd-IgA1 consists of terminal N-acetyl-galactosamine (GalNAc) or sialylated GalNAc [20–23]. These studies identified excess Gd-IgA1 in both serum and glomerular immune deposits in patients with IgAN [20–23]. Furthermore, the recently proposed multi-hit theory of IgAN states that overproduced Gd-IgA1 and autoantibodies against Gd-IgA1 subsequently form circulating IC, resulting in glomerular mesangial deposits followed by accelerated nephritis [20, 24]. Thus, Gd-IgA1 is vital to the pathogenesis of IgAN.

Intriguingly, HSPN and IgAN have been described consecutively in the same patient and in identical twins [14]. Moreover, recent studies noted that these 2 diseases share the feature of Gd-IgA1-oriented pathogenesis [25, 26]. In our recent report, serum Gd-IgA1 (s-Gd-IgA1) levels, quantified by a novel lectin-independent enzyme-linked immunosorbent assay (ELISA) using an anti-Gd-IgA1 monoclonal antibody (KM55) [27], were significantly elevated in patients with HSPN or IgAN compared to other kidney diseases [26]. Similar results were obtained in other recent studies [27, 28]. Additionally, glomerular-Gd-IgA1 (g-Gd-IgA1)-

deposition, assessed by immunofluorescence (IF) or immunohistochemistry (IHC) with KM55, was specific to patients with HSPN and IgAN [25, 26]. Taken together, HSPN and IgAN have similar pathologic and biological abnormalities and are closely related.

However, there are also distinct clinical differences between the 2 diseases. HSPN can be regarded as a renal symptom of systemic vasculitis. Patients with HSPN present palpable purpura and gastrointestinal bleeding or polyarthritis other than GN [2]. Meanwhile, IgAN is restricted to the kidneys. Furthermore, symptoms in the acute phase of HSP are sometimes self-limiting and resolve without intensive treatment. HSPN tends to have a good prognosis compared to IgAN [10, 13]. Therefore, it is interesting to explore whether IgAN could be recognized as renal-limited HSP and if pathogenic mechanisms proposed for IgAN may also apply to HSPN. In this study, we aimed to clarify the clinicopathologic dissimilarities between HSPN and IgAN to address these clinical questions.

## Patients and methods

### Study design and participants

We performed a cross-sectional study of 80 adult (≥18 years) patients with HSPN (n = 24) or IgAN (n = 56) who underwent a renal biopsy (RB) between April 2008 and December 2018 at Showa University Hospital. Data for some patients with HSPN or IgAN were previously described in our historical cohort study [26]. Furthermore, 6 patients with lupus nephritis (LN), 10 with ANCA-associated vasculitis (AAV), and 6 with minimal change disease (MCD) were enrolled as positive or negative controls for ELISA measurements.

HSPN was diagnosed according to a modification of the European League Against Rheumatism/the Pediatric Rheumatology International Trials Organization/the Pediatric Rheumatology European Society (EULAR/PRINTO/PRES) classification criteria [29]: purpura or petechiae with lower limb predominance and the presence of urinary abnormalities, renal insufficiency, and predominant mesangial IgA deposits on RB. All but 2 patients underwent a skin biopsy and were confirmed to have leukocytoclastic vasculitis with predominant IgA deposits. Purpura had developed before nephritis or at the same time as nephritis in all patients with HSPN.

All patients provided written informed consent regarding preservation of blood samples, urine samples, and kidney tissues. In addition, they provided written informed consent to make all data obtained from RB available. Opt-out methods were used to obtain informed consent regarding the measurement of s-Gd-IgA1 values and evaluation of g-Gd-IgA1 deposition. All enrolled patients agreed to participate in this study. The Ethics Committee at Showa University Hospital approved the study protocol (No. 2831), and the study proceeded in accordance with the ethical standards of the Declaration of Helsinki.

### Clinical and pathologic parameters

Clinical characteristics, including information on duration from onset of abnormal urinalysis findings to the time of RB (duration from onset), age, sex, body mass index (BMI), history of hypertension (HT), mean arterial pressure (MAP), absence of complications such as gastrointestinal bleeding and polyarthritis, absence of hematuria, degree of urinary protein, urinary N-acetyl-beta-D-glucosaminidase (NAG) index, serum creatinine (sCr), estimated glomerular filtration rate (eGFR), serum IgA and C3, and therapeutic regimens were obtained from patient records. A blood pressure ≥135/85 mmHg was defined as HT. We calculated MAP according to previous report [30]. Hematuria was scored from 0 to 3+ as described previously [31]. We calculated eGFR using the modified Modification of Diet in Renal Disease (MDRD) equation for Japanese persons [32]. Therapeutic regimens at the point of RB, including information

about treatment with renin-angiotensin system inhibitors (RASI) and steroids, were assessed. In addition, implementation of steroid pulse therapy combined with tonsillectomy (TSP) after RB was also analyzed. Indications for and the regimen for TSP in IgAN are detailed elsewhere [24, 33, 34].

Histologic sections were independently reviewed by 2 renal pathologists who were blinded to the clinical data of patients. The International Study of Kidney Disease in Children (ISKDC) classification based on the degree of mesangial proliferation and the rate of crescent formation was used to analyze the histologic severity of HSPN [35]. The histologic severity of IgAN was determined according to the histologic grading criteria of the Japanese Society of Nephrology (JSN) [36]. The Oxford classification was also used to evaluate or categorize histologic findings [37].

## Immunofluorescence staining

Two nephrologists independently analyzed mesangial IgA, IgG, IgM, and fibrin deposits using IF staining according to our protocol [38] and graded the IF intensity of mesangial IgA as described elsewhere [31].

## ELISA for s-Gd-IgA1

Levels of s-Gd-IgA1 were measured using sandwich ELISA kits with KM55 (#27600, Immuno-Biological Laboratories, Fujioka, Japan) [27]. Serum samples were diluted with EIA buffer (1:800), and levels were measured as recommended by the manufacturer.

## ELISA for inflammatory cytokines

Levels of serum interleukin (IL)-8, monocyte chemoattractant protein-1 (MCP-1), IL-6, and tumor necrosis factor-$\alpha$ (TNF-$\alpha$) were measured using specific sandwich ELISA kits from R&D Systems (IL-8, #D8000C; MCP-1, #DCP00; IL-6, #D6050, TNF-$\alpha$, #DTA00D; Abingdon, U.K.). Serum samples, except for MCP-1, were measured without dilution, and levels were measured as recommended by the manufacturer. Samples for measuring MCP-1 levels were diluted with RD6Q buffer (x 2).

## Immunohistochemistry for IgA

IgA deposition in the glomerulus was identified by IHC staining with an IgA-detection kit (BioGenex, Hague, Netherlands) as previously described [39, 40]. Briefly, the paraffin sections of tissues were dewaxed and washed in phosphate buffered saline (PBS). $H_2O_2$ (0.3%) in methanol was added to slides for 30 min to quench endogenous peroxidase. Sections were washed in PBS and incubated for 60 min at room temperature with rabbit anti-human IgA polyclonal antibody (#AR045-5R, #PU045-UP, 1:500 antibody dilution). Sections were washed 3 times in PBS, then incubated with EnVisionTM+Dual Link System -HRP (Dako, Glostrup, Denmark) for 60 min at room temperature. Next, the sections were developed using diaminobenzidine (DAB) (Dako) as the substrate to produce a brown stain, and sections were counterstained with hematoxylin.

The intensity of IgA in glomerular areas was assessed as 0, none; 1, mild; 2, moderate; or 3, severe. Two nephrologists independently scored IgA intensity in all glomeruli in each section under × 400 magnification, and the mean value per glomerulus of each section was determined. Also, the intensity of IgA in glomerular mesangial or endothelial areas was similarly assessed.

## Immunohistochemistry for Gd-IgA1

Gd-IgA1 deposition in glomeruli was examined by IHC staining as described elsewhere [26, 39, 40]. Briefly, dewaxed paraffin sections were heated with Histofine (Nichirei, Tokyo, Japan) in an autoclave at 121˚C for 30 min for antigen retrieval. After endogenous peroxidase was quenched with 0.3% $H_2O_2$ in methanol, nonspecific binding was blocked with protein blocking solution, and sections were incubated overnight at 4˚C with rat monoclonal anti-human Gd-IgA1 antibody (KM55) (#10777, Immuno-Biological Laboratories) diluted to 1:100, followed by EnVisionTM+Dual Link System-HRP (Dako) for 60 min at room temperature. Color was then developed using DAB (Dako).

The intensity of Gd-IgA1 in glomerular areas was assessed as 0, none; 1, mild; 2, moderate; or 3, severe. Two nephrologists independently scored g-Gd-IgA1 intensity in all glomeruli in each section under × 400 magnification, and the mean value per glomerulus of each section was determined. Also, the intensity of Gd-IgA1 in glomerular mesangial or endothelial areas was similarly assessed.

## Double immunostaining for CD31 with Gd-IgA1

Two-color immunostaining was used to detect colocalization of CD31, an endothelial cell surface marker [41], with Gd-IgA1 according to our protocol [42]. Briefly, as described above, paraffin sections were heated with Histofine in an autoclave at 121˚C for antigen retrieval. Thereafter, sections were stained with KM-55 (1:100 dilution) and rabbit anti–human CD31 Ab (Abcam, Cambridge, United Kingdom, 1:200 dilution), followed by EnVisionTM+Dual Link System-HRP (Dako) and EnVisionTM G/2 System/AP (Dako). Color was then developed using DAB (Dako) or permanent red (Dako).

## Statistical analysis

Data are expressed as means ± SD or SEM or ratios (%). Results were analyzed using Prism software (GraphPad Software Inc., La Jolla, CA, USA). Non-parametric variables were compared using either Mann-Whitney U tests or Kruskal-Wallis tests. Categorical variables were compared using Fisher exact tests. Correlations between parameters were assessed using Spearman correlation coefficients. P values of $< 0.05$ were considered to be statistically significant in all the analyses.

# Results

## Clinical characteristics

Table 1 summarizes the outcomes of comparisons of clinical characteristics at the time of RB and the therapeutic regimens between 24 patients with HSPN (male, 13; mean age [± SD], 44.4 ± 19.3 years) and 56 patients with IgAN (male, 26; 37.4 ± 13.9 years). The clinical characteristics did not significantly differ between groups except for duration from onset and degree of hematuria. Compared to patients with IgAN, patients with HSPN underwent RB significantly earlier after onset of urinary abnormality. In terms of renal function, the mean (± SD) for sCr level was lower, and eGFR level tended to be higher in patients with HSPN compared to IgAN, although differences were not significant. Regarding therapeutic regimens, significantly more patients with HSPN received steroid therapy (ST) than patients with IgAN. In HSPN, 9 of 24 patients had already received ST against purpura rather than GN at the point of RB. In addition, significantly more patients with IgAN were treated with RASI and TSP than patients with HSPN.

**Table 1. Clinical characteristics of patients with HSPN and IgAN.**

|  | HSPN | IgAN | P value |
|---|---|---|---|
| **Characteristics** | **(n = 24)** | **(n = 56)** |  |
| Age (years) | 44.4 ± 19.3 | 37.1 ± 13.9 | 0.174 |
| Male gender, No. (%) | 13 (54.2%) | 26 (46.4.%) | 0.525 |
| Duration from onset (months) | 8.8 ± 11.8 | 47.6 ± 65.1 | <**0.001** |
| BMI (kg/m$^2$) | 23.0 ± 5.2 | 22.4 ± 4.3 | 0.741 |
| History of HT[a], No. (%) | 3 (12.5%) | 14 (25.0%) | 0.210 |
| MAP[b] (mmHg) | 88.7 ± 13.3 | 89.4 ± 11.8 | 0.603 |
| Proteinuria (g/day) | 1.3 ± 2.1 | 1.2 ± 1.4 | 0.305 |
| Hematuria[c], No. (%) (±) | 2 (8.3%) | 2 (3.6%) | 0.370 |
| (1+) | 11 (45.8%) | 7 (12.5%) | **0.001** |
| (2+) | 5 (20.8%) | 11 (19.6%) | 0.903 |
| (3+) | 6 (25.0%) | 36 (64.3%) | **0.001** |
| Urinary NAG index (U/gCr) | 10.7 ± 9.9 | 11.1 ± 10.0 | 0.531 |
| sCr (mg/dL) | 0.7 ± 0.2 | 1.0 ± 0.6 | 0.053 |
| eGFR (mL/min/1.73 mm$^2$) | 81.6 ± 19.0 | 74.4 ± 31.2 | 0.282 |
| Alb (g/dL) | 3.8 ± 0.6 | 3.9 ± 0.6 | 0.373 |
| Serum IgA (mg/dL) | 326.0 ± 109.7 | 330.9 ± 108.8 | 0.829 |
| IgA/C3 | 3.0 ± 1.3 | 3.3 ± 1.3 | 0.364 |
| Treatment, No. (%) |  |  |  |
| Use of RASI at RB | 1 (4.2%) | 15 (26.8%) | **0.020** |
| Steroid therapy at RB | 9 (37.5%) | 1 (1.8%) | <**0.001** |
| Underwent TSP after RB | 1 (4.2%) | 13 (23.2%) | **0.039** |

Value are means ± SD or (percent). Mann-Whitney U test or Fisher's test were used for statistical analysis.

Abbreviations: HSPN, Henoch-Schönlein purpura nephritis; IgAN, Immunoglobulin A nephropathy; BMI, body mass index; HT, hypertension, MAP, mean arterial pressure; sCr, serum creatinine; eGFR, estimated glomerular filtration rate; Alb, albumin; RB, renal biopsy; RASI, renin-angiotensin system inhibitor; TSP, steroid pulse therapy combined with tonsillectomy.

[a]Blood pressure ≥135/85 mmHg was defined as hypertension.

[b]MAP was calculated according to previous reports [30].

[c]Hematuria was scored from 0 to 3+ as described previously [31].

## Histologic findings

Table 2 summarizes the comparisons of histologic findings between 24 patients with HSPN and 56 patients with IgAN. The proportions of grade I, II, IIIa, IIIb, and IV in ISKDC classification for HSPN were 0%, 16.7%, 83.3%, 0%, and 0%, respectively. The proportions of histological-grade (H-grade) I, II, III, and IV using the JSN classification for IgAN were 32.1%, 33.9%, 33.9%, and 0%, respectively. Compared to IgAN, the mean rate (± SD) of global sclerosis or crescent formation was significantly lower in patients with HSPN. When applying the histologic severity based on Oxford classification to HSPN, endothelial injury was significant in patients with HSPN, although mesangial proliferation, crescent formation, and tubulointerstitial injury were significantly worse in patients with IgAN. With regard to glomerular IC deposition on IF or electron microscope assessment, patients with HSPN showed a significantly higher rate of fibrin deposition and electron dense deposition in the subendothelial area compared to patients with IgAN, although there was no significant difference in the mesangial IgA or IgG depositions between the 2 diseases.

**Table 2. Comparison of histologic findings between patients with HSPN and IgAN.**

| | HSPN | IgAN | P value |
|---|---|---|---|
| **Characteristics** | **(n = 24)** | **(n = 56)** | |
| ISKDC classification[a], No (%) | | | |
| I | 0 (0%) | | |
| II | 4 (16.7%) | | |
| IIIa | 20 (83.3%) | | |
| IIIb | 0 (0%) | | |
| IV | 0 (0%) | | |
| Histological grade according to JSN[b], No (%) | | | |
| I | | 18 (32.1%) | |
| II | | 19 (33.9%) | |
| III | | 19 (33.9%) | |
| IV | | 0 (0%) | |
| Global sclerosis rate[c] (%) | 7.3 ± 10.3 | 19.4 ± 17.2 | **0.002** |
| Crescent rate[c] (%) | 5.4 ± 8.0 | 16.2 ± 18.2 | **0.006** |
| Global sclerosis + crescent rate[c] (%) | 14.4 ± 21.5 | 35.8 ± 20.9 | **<0.001** |
| Oxford classification[d], No (%) | | | |
| M1 | 12 (50.0%) | 45 (80.3%) | **0.006** |
| E1 | 18 (75.0%) | 13 (23.2%) | **<0.001** |
| S1 | 10 (41.6%) | 16 (28.6%) | 0.195 |
| T1-2 | 0 (0%) | 15 (26.8%) | **0.003** |
| C1-2 | 12 (50%) | 41 (73.2%) | **0.044** |
| Glomerular deposition on IF staining | | | |
| IgA: weak[e] | 2 (8.3%) | 10 (17.8%) | 0.331 |
| IgA: moderate[e] | 10 (41.7%) | 29 (51.8%) | 0.407 |
| IgA: strong[e] | 12 (50.0%) | 17 (30.4%) | 0.094 |
| IgG deposition | 9 (37.5%) | 15 (26.8%) | 0.489 |
| Fibrin deposition, | 21 (87.5%) | 6 (10.7%) | **<0.001** |
| Electron dense deposit, No (%) | | | |
| Mesangial area | 23 (95.8%) | 55 (98.2%) | 0.087 |
| Subendothelial area | 11 (45.8%) | 3 (5.4%) | **<0.001** |

Value are means ± SD or (percent). Mann-Whitney U test or Fisher's test are used for statistical analysis.

Abbreviations: HSPN; Henoch-Schönlein purpura nephritis, IgAN; Immunoglobulin A nephropathy; ISKDC, International Study of Kidney Disease in Children; JSN, Japanese Society of Nephrology, IF; immunofluorescence.

[a]Histologic classification in HSPN was graded based on the ISKDC classification [35].

[b]Histological grade in IgAN was classified according to the criteria of the JSN [36].

[c]Rates of global sclerosis, crescents, and both types of glomerular lesions (%) were calculated by dividing total number of each type of lesion by total number of glomeruli. Crescents comprise cellular, fibrocellular, and fibrous types.

[d]Histological severity was graded according to Oxford classification [37].

[e]Intensity of IgA deposition was described earlier [31].

## Levels of s-Gd-IgA1 and intensity of g-Gd-IgA1 deposition

Fig 1A shows that mean (± SD) s-Gd-IgA1 levels were significantly elevated in patients with HSPN compared to patients with MCD (HSPN vs MCD: 13.1 ± 9.7 vs. 5.2 ± 1.8 μg/mL, p = 0.005). In addition, s-Gd-IgA1 values were significantly elevated in patients with IgAN compared to patients with MCD (16.1 ± 10.2 vs. 5.2 ± 1.8 μg/mL, p = 0.001), but did not significantly differ between patients with IgAN or HSPN (Fig 1A). Even after correction for sCr levels, s-Gd-

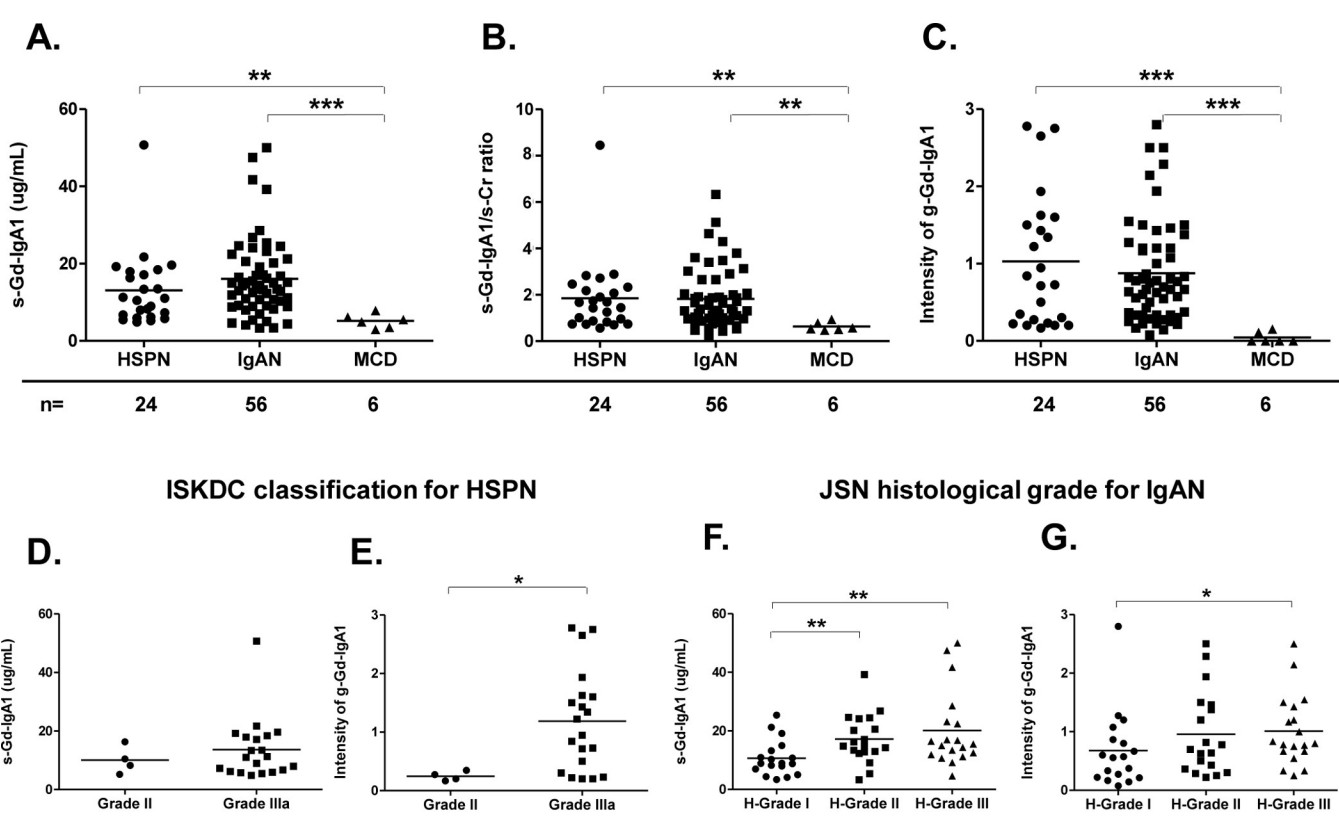

**Fig 1. S-Gd-IgA1 levels determined by ELISA and g-Gd-IgA1 deposition identified by IHC staining between HSPN and IgAN.** S-Gd-IgA1 levels in patients with HSPN, IgAN, or MCD (**A**). Levels of s-Gd-IgA1 after correction for sCr levels. S-Gd-IgA1 values divided by sCr values for individual patients and compared among the study groups (**B**). Intensity of g-Gd-IgA1 among patients with HSPN, IgAN, or MCD (**C**). S-Gd-IgA1 level or m-Gd-IgA1 intensity in patients with HSPN according to ISKDC classification (**D** and **E**). S-Gd-IgA1 level or g-Gd-IgA1 intensity in patients with IgAN according to JSN classification (**F** and **G**). Horizontal solid lines represent means. Data were statistically analyzed using Kruskal-Wallis tests and Mann-Whitney U tests. $^{*}P<0.05$, $^{**}P<0.01$, and $^{***}P<0.001$.

IgA1 levels were also significantly higher in patients with HSPN or IgAN than in patients with MCD (Fig 1B). Glomerular Gd-IgA1 deposition was apparently specific to HSPN and IgAN (Fig 1C) at significantly higher intensity, and mean (± SD) values for g-Gd-IgA1 staining were significantly higher in these patients than in patients with MCD (HSPN vs. MCD: 1.0 ± 0.9 vs. 0.1 ± 0.1, p < 0.001) (IgAN vs. MCD: 0.9 ± 0.7 vs. 0.1 ± 0.1, p < 0.001) (Fig 1C).

Among patients with HSPN categorized using the ISKDC classification based on the degree of mesangial proliferation and the presence of crescents, s-Gd-IgA1 levels tended to be higher in those with grade IIIa disease than grade II disease, although differences were not significant (10.1 ± 4.7 vs. 13.7 ± 10.4 µg/mL, p = 0.255) (Fig 1D). Staining for g-Gd-IgA1 was significantly more intense among patients with grade IIIa than II disease (0.2 ± 0.1 vs. 1.2 ± 0.9, p = 0.022) (Fig 1E). Similarly, among patients with IgAN categorized as JSN H-grade based on the ratio of global sclerosis, segmental sclerosis, and crescents, s-Gd-IgA1 levels were significantly higher in those with grade II or III disease than grade I disease (17.2 ± 8.3 or 20.1 ± 12.9 vs. 10.7 ± 6.1 µg/mL, p = 0.008 and p = 0.003, respectively) (Fig 1F). Staining for g-Gd-IgA1 was significantly more intense among patients with H-grades III than I according to the JSN classification (1.0 ± 0.6 vs. 0.6 ± 0.6, p = 0.037) (Fig 1G). Additionally, g-Gd-IgA1 intensity scores positively correlated with s-Gd-IgA1 values in patients with IgAN (r = 0.082, p = 0.032) (S1B Fig), whereas no correlation was detected in patients with HSPN (S1A Fig).

Furthermore, we focused on the influence of ST on both types of Gd-IgA1 in patients with HSPN. S2 Fig shows levels of s-Gd-IgA1 and intensity of g-Gd-IgA1 deposition among HSPN patients who received ST [HSPN-ST (+), n = 9] and HSPN patients who did not receive ST [HSPN-ST (-), n = 15] at the point of RB. As shown in S2A Fig, s-Gd-IgA1 levels tended to be higher in HSPN-ST (+) compared to HSPN-ST (-) although the differences were not statistically significant. Values of g-Gd-IgA1 positivity were comparable between two groups (S2B Fig). In addition, similar to the results in S1A Fig, no correlation between s-Gd-IgA1 levels and g-Gd-IgA1 intensity was detected with either HSPN-ST (+) (S2C Fig) or HSPN-ST (-) (S2D Fig).

### Levels of serum inflammatory cytokines

Fig 2 shows the levels of serum inflammatory cytokines including IL-8, MCP-1, TNF-α, and IL-6 among patients with HSPN, IgAN, AAV, LN, and MCD. Levels of serum IL-8 in patients with HSPN or IgAN were significantly higher than those in patients with LN or MCD. Mean (± SEM) IL-8 levels in patients with HSPN were significantly higher than levels in patients with IgAN (62.8 ± 24.9 vs. 21.2 ± 4.4 pg/mL, p = 0.033) (Fig 2A), and the elevation of IL-8 in patients with HSPN was comparable with patients with AAV (Fig 2A). In addition, levels of

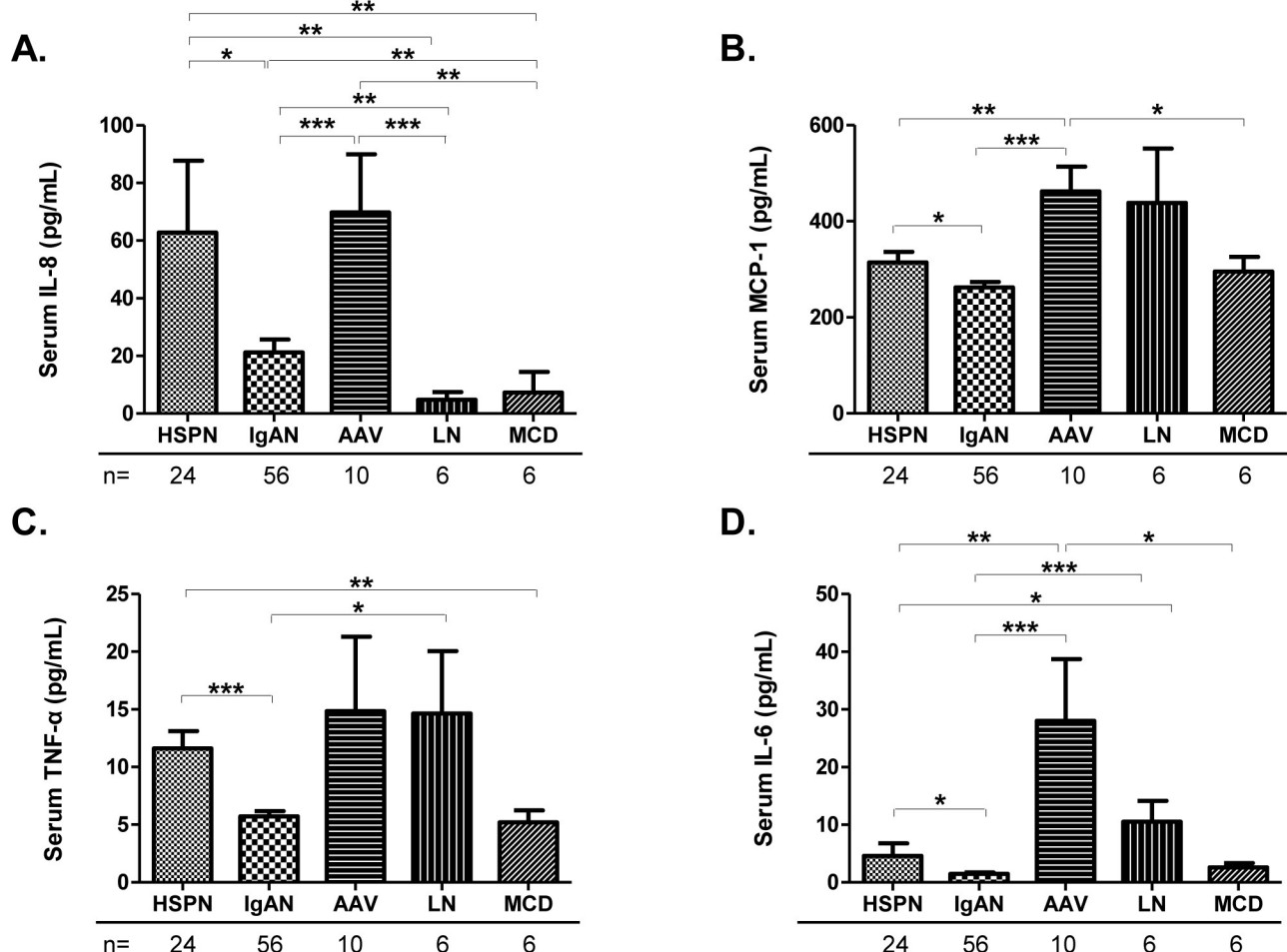

**Fig 2. Serum inflammatory cytokines levels determined by ELISA between patients with HSPN or IgAN.** Levels for serum IL-8 (**A**), MCP-1 (**B**), TNF-α (**C**), and IL-6 (**D**) among patients with HSPN, IgAN, AAV, LN, or MCD. Values are presented as means ± SEM. Data were statistically analyzed using Kruskal-Wallis tests and Mann-Whitney U tests. $^*P<0.05$, $^{**}P<0.01$, and $^{***}P<0.001$.

serum TNF-α in patients with HSPN were significantly higher than levels in patients with MCD (Fig 2C). Patients with HSPN showed significant elevations of TNF-α (mean ± SEM) compared to patients with IgAN (11.6 ± 1.5 vs. 5.8 ± 0.5 pg/mL, p<0.001) (Fig 2C).

Levels of MCP-1 and IL-6 in patients with HSPN or IgAN were not elevated and were significantly lower compared to patients with AAV or LN. However, serum concentrations (mean ± SEM) of those 2 cytokines were higher in patients with HSPN than in patients with IgAN (MCP-1: 313.5 ± 21.0 vs. 262.4 ± 11.4 pg/mL, p = 0.022) (Fig 2B), (IL-6: 4.6 ± 2.2 vs. 1.5 ± 0.3 pg/mL, p = 0.045) (Fig 2D).

To assess the influence of ST at the point of RB on serum inflammatory cytokines in HSPN, we compared levels of inflammatory cytokines among HSPN-ST (+) and HSPN-ST (-). As shown in S3A Fig, serum concentrations (mean ± SEM) of IL-8 were significantly higher in HSPN-ST (-) compared to HSPN-ST (+) (86.5 ± 38.1 vs. 23.4 ± 13.9 pg/mL, p = 0.034). Serum levels for MCP-1, TNF-α, and IL-6 were not significantly different between the two groups (S3B, S3C and S3D Fig).

## Association of Gd-IgA1 with laboratory parameters or pathologic findings

Table 3 summarizes associations between both types of Gd-IgA1 and laboratory parameters or pathologic findings in 24 patients with HSPN and 56 patients with IgAN. Serum IgA and IgA/C3 ratios positively correlated with s-Gd-IgA1 in patients with IgAN but not in patients with HSPN (Table 3).

In terms of renal function, g-Gd-IgA1 significantly correlated with sCr values in patients with HSPN (Table 3). S-Gd-IgA1 negatively correlated with eGFR values in patients with IgAN (Table 3). Furthermore, s-Gd-IgA1 significantly correlated with global sclerosis plus crescent rates in patients with HSPN or IgAN. Neither type of Gd-IgA1 correlated with proteinuria or urinary NAG index in either disease. Of note, s-Gd-IgA1 in patients with HSPN significantly correlated with crescent rates, whereas s-Gd-IgA1 in patients with IgAN significantly correlated with global sclerosis rates (Table 3).

Based on the Oxford classification, s-Gd-IgA1 levels were significantly higher in patients with IgAN with segmental sclerosis and tubular atrophy/interstitial fibrosis (S4B Fig), and g-

**Table 3. Correlation between both types of Gd-IgA1 and clinicopathologic parameters in patients with HSPN or IgAN.**

| | HSPN (n = 24) | | | | IgAN (n = 56) | | | |
| | s-Gd-IgA1 (µg/mL) | | g-Gd-IgA1 intensity | | s-Gd-IgA1 (µg/mL) | | g-Gd-IgA1 intensity | |
| Variable | R value | P value | R value | P value | R value | P value | R value | P value |
|---|---|---|---|---|---|---|---|---|
| Serum IgA (mg/dL) | 0.014 | 0.580 | 0.019 | 0.517 | 0.429 | **<0.001** | 0.049 | 0.102 |
| IgA/C3 | <0.001 | 0.884 | 0.004 | 0.766 | 0.276 | **<0.001** | 0.060 | 0.068 |
| Proteinuria (g/day) | 0.061 | 0.254 | 0.124 | 0.099 | 0.003 | 0.666 | 0.002 | 0.766 |
| Serum Cr (mg/dL) | 0.013 | 0.595 | 0.249 | **0.013** | 0.056 | 0.080 | 0.006 | 0.559 |
| eGFR (mL/min/1.73 mm²) | 0.012 | 0.616 | 0.014 | 0.583 | -0.092 | **0.023** | 0.002 | 0.757 |
| NAG (U/gCr) | 0.002 | 0.856 | 0.047 | 0.319 | 0.008 | 0.523 | 0.014 | 0.399 |
| Global sclerosis rate[a] (%) | <0.001 | 0.944 | 0.011 | 0.621 | 0.207 | **<0.001** | 0.026 | 0.233 |
| Crescent rate[a] (%) | 0.454 | **<0.001** | 0.003 | 0.811 | 0.008 | 0.522 | <0.001 | 0.831 |
| Global sclerosis + Crescent rate[a] (%) | 0.632 | **<0.001** | <0.001 | 0.929 | 0.088 | **0.026** | 0.029 | 0.206 |

Data were statistically analyzed using Spearman correlation tests.

Abbreviations: HSPN, Henoch-Schönlein purpura nephritis; IgAN, Immunoglobulin A nephtopathy; Cr, creatinine; eGFR, estimated glomerular filtration rate.

[a]Rates of global sclerosis, crescents, and both types of glomerular lesions (%) were calculated by dividing the total number of each type of lesion by the total number of glomeruli. Crescents comprise cellular, fibrocellular, and fibrous types.

Gd-IgA1 intensity in patients with IgAN was significantly higher in patients with segmental sclerosis, tubular lesions, and crescent formation (S4D Fig). On the other hand, based on the Oxford classification of patients with HSPN, neither type of Gd-IgA1 showed any significant difference among classifications (S4A and S4C Fig), although s-Gd-IgA1 showed a tendency to be higher in patients with crescent formation (S4A Fig).

## Association of Gd-IgA1 with inflammatory cytokines

Table 4 shows associations between serum inflammatory cytokines and both types of Gd-IgA1 in patients with HSPN and IgAN. Although patients with IgAN showed no correlation of inflammatory cytokines with Gd-IgA1, patients with HSPN showed a significant positive correlation between s-Gd-IgA1 and IL-6 (Table 4). In addition, g-Gd-IgA1 intensity in patients with HSPN was positively correlated with IL-8 and MCP-1 elevations (Table 4).

We also evaluated those associations with HSPN-ST (+) or HSPN-ST (-). As shown in S1 Table, serum IL-6 levels showed a significant positive correlation with s-Gd-IgA1 levels in HSPN-ST (+) and a significant positive correlation with g-Gd-IgA1 intensity in HSPN-ST (-). On the other hand, other serum cytokines, including IL-8, MCP-1, and TNF-α, did not significantly correlate with either type of Gd-IgA1 in either group (S1 Table), However those 3 cytokines showed a tendency to positively correlate with g-Gd-IgA1 intensity in HSPN-ST (-) (S1 Table).

## Association of inflammatory cytokines with HSPN progression

To clarify the association between inflammatory cytokines and HSPN severity, we compared levels of inflammatory cytokines determined by ELISA among patients with HSPN. S5 Fig shows levels of IL-8, MCP-1, TNF-α, and IL-6 between patients with HSPN without any systemic symptoms other than nephritis and patients with HSPN with arthritis or abdominal symptoms (HSPN-AA). As shown in S5A–S5D Fig, serum levels for IL-8, MCP-1, and IL-6 tended to be higher in patients with HSPN-AA compared to patients with HSPN only although none of the differences showed statistical significance.

We also compared serum inflammatory cytokines among groups based on the Oxford classification (Fig 3A–3H). The presence of mesangial hypercellularity, segmental sclerosis, and interstitial fibrosis/tubular atrophy lesions did not all lead to significant differences in levels of serum inflammatory cytokines among patients with HSPN (S6A–S6L Fig). However, mean (± SEM) IL-8 levels were higher in patients with HSPN with endocapillary lesions (75.6 ± 32.5 vs. 24.5 ± 14.9 pg/mL, p = 0.067) (Fig 3A). Also, mean (± SEM) MCP-1 levels were significantly elevated in patients with HSPN with crescent formation compared to the patients without crescent formation (352.6 ± 38.1 vs. 275.8 ± 16.7 pg/mL, p = 0.039) (Fig 3F).

**Table 4. Correlation between both types of Gd-IgA1 and inflammatory cytokines in patients with HSPN and IgAN.**

| | HSPN (n = 24) | | | | IgAN (n = 56) | | | |
| | s-Gd-IgA1 (μg/mL) | | g-Gd-IgA1 intensity | | s-Gd-IgA1 (μg/mL) | | g-Gd-IgA1 intensity | |
| Variable | R value | P value | R value | P value | R value | P value | R value | P value |
|---|---|---|---|---|---|---|---|---|
| IL-8 (pg/mL) | 0.079 | 0.184 | 0.165 | 0.052 | 0.049 | 0.100 | 0.002 | 0.727 |
| MCP-1 (pg/mL) | 0.003 | 0.771 | 0.197 | **0.030** | 0.012 | 0.423 | 0.056 | 0.079 |
| TNF-α (pg/mL) | 0.049 | 0.297 | 0.001 | 0.902 | 0.037 | 0.157 | 0.001 | 0.901 |
| IL-6 (pg/mL) | 0.487 | <**0.001** | 0.068 | 0.218 | 0.001 | 0.506 | 0.024 | 0.250 |

Data were statistically analyzed using Spearman correlation tests.

Abbreviations: HSPN, Henoch-Schönlein purpura nephritis; IgAN, Immunoglobulin A nephtopathy.

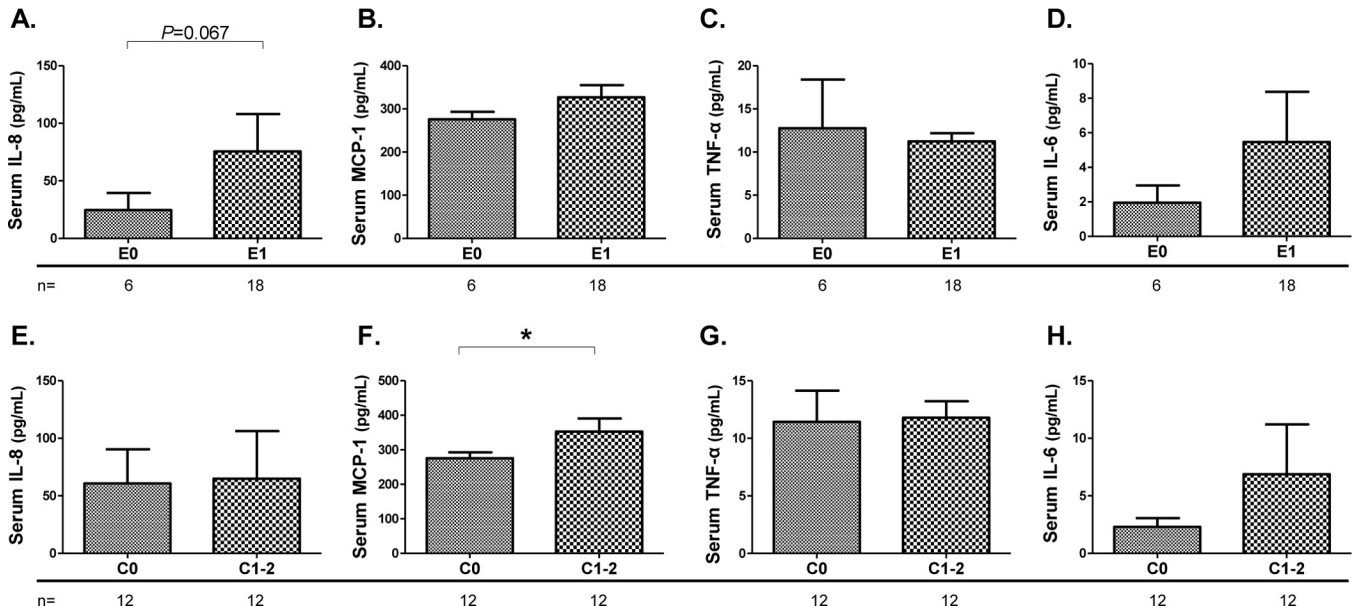

**Fig 3. Comparisons of serum inflammatory cytokines among groups based on the Oxford classification of patients with HSPN.** Comparison of serum IL-8 (**A** and **E**), MCP-1 (**B** and **F**), TNF-α (**C** and **G**), and IL-6 (**D** and **H**) in patients with HSPN according to the presence of endothelial lesions or crescent formation based on the Oxford classification. Values are presented as means ± SEM. Data were statistically analyzed using Mann-Whitney U tests. *$P<0.05$.

## Distinction between glomerular IgA and Gd-IgA1 deposition in HSPN vs. IgAN

We focused on the distribution of glomerular IgA and Gd-IgA1 deposition between patients with HSPN and IgAN. Consecutive paraffin-embedded sections of RB specimens, obtained form 24 patients with HSPN and 56 patients with IgAN, were stained with anti-human IgA polyclonal antibody and KM-55. Fig 4A–4D shows representative photos of IHC staining for IgA and Gd-IgA1 in the glomerulus. As shown in Fig 4A and 4B, intensity of mesangial IgA deposition in patients with HSPN was comparable with that of patients with IgAN, whereas the mean (± SEM) intensity of endothelial IgA deposition was significantly higher in patients with HSPN ($1.1 ± 0.1$ vs. $0.8 ± 0.1$, $p < 0.001$) (Fig 4E).

Although the intensity of g-Gd-IgA1 deposition was generally lower than that of IgA deposition in patients with HSPN or IgAN (Fig 4C and 4D), both diseases showed a similar degree of Gd-IgA1 deposition in the mesangial area (Fig 4F). Regarding endothelial Gd-IgA1 deposition, both diseases showed weak staining compared to the Gd-IgA1 positivity in the mesangial area, although patients with HSPN showed a statistically higher degree of endothelial Gd-IgA1 deposition (Fig 4F).

Furthermore, 2-color immunohistochemistry with CD31 stained red and Gd-IgA1 stained brown was performed to confirm that g-Gd-IgA1 deposition is dominant in the mesangial area compared with the endothelial area. As shown in Fig 5A and 5B, positivity of Gd-IgA1 was significant in the glomerular mesangial region in both diseases. Most Gd-IgA1-positive mesangial areas were not double stained with CD31 in HSPN and IgAN samples, although partial double-stained areas were occasionally seen in HSPN samples (Fig 5C and 5D).

## Discussion

In the present study, we compared detailed clinicopathologic information, including s-Gd-IgA1 levels, g-Gd-IgA1deposition, and levels of several serum inflammatory cytokines between

**Fig. 4**

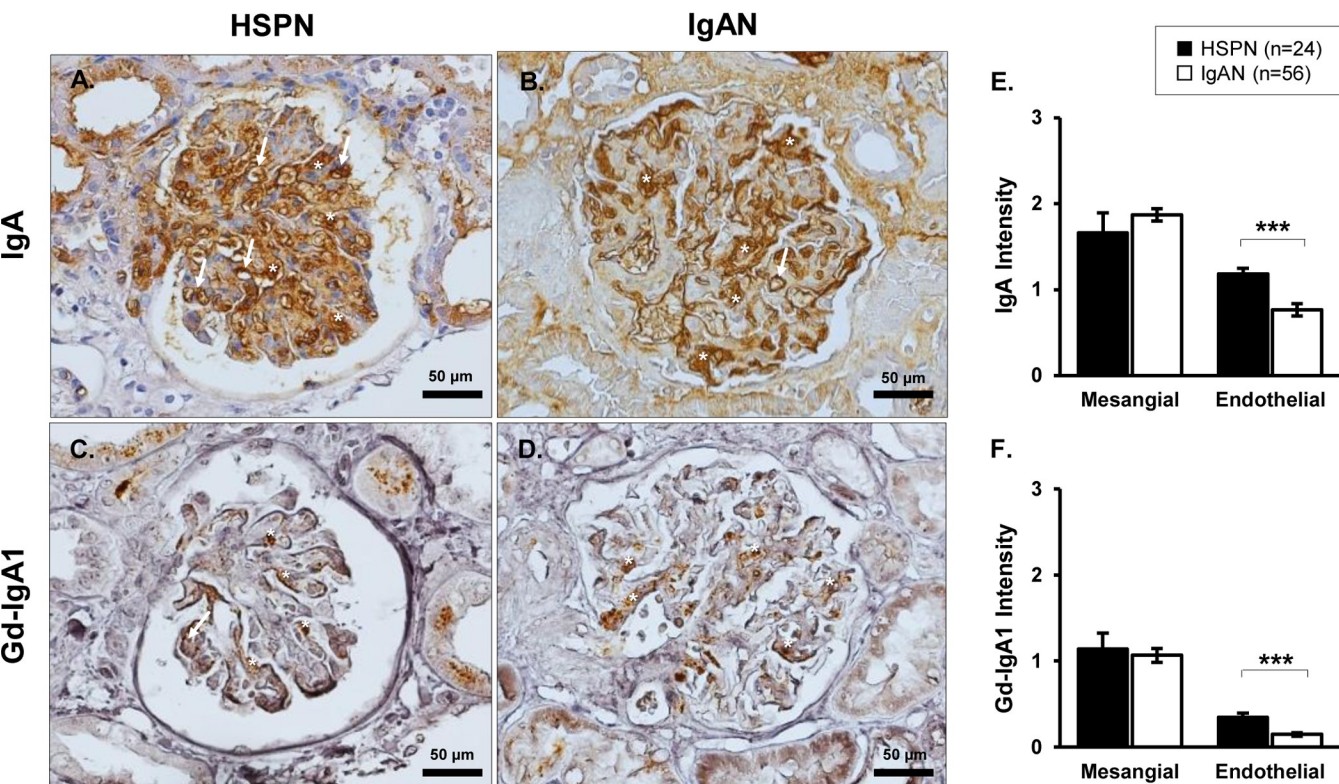

**Fig 4. Distinction between glomerular IgA and Gd-IgA1 deposition identified by IHC staining in HSPN vs. IgAN.** Consecutive paraffin-embedded sections, obtained form 24 patients with HSPN and 56 patients with IgAN, were stained with anti-human IgA polyclonal antibody and KM-55. Representative photos of IgA staining (brown reaction product) (**A** and **B**) and Gd-IgA1 (brown reaction product) staining (**C** and **D**). Endothelial positive areas are shown by white arrows, and mesangial positive areas were shown by white asterisks. Original magnification: ×40. Intensity of mesangial or endothelial IgA deposition among patients with HSPN or IgAN (**E**). Intensity of mesangial or endothelial Gd-IgA1 deposition among patients with HSPN or IgAN (**F**). Values are presented as the mean ± SEM. Data were statistically analyzed using Mann-Whitney U tests. ***$P<0.001$.

patients with HSPN or IgAN, which led to clarify similarities and differences between these 2 immuno-pathologically indistinguishable diseases. Therefore, our results might provide suggestive information to physicians.

Based on our findings, patients with HSPN underwent RB at a relatively early phase after the onset of GN and a greater number of patients received ST. In contrast, patients with IgAN tended to received supportive therapy such as RASI before diagnosis and undergo RB at relative late phase after the onset of GN. Consequently, histological grade in patients with HSPN was mild to moderate. Meanwhile RB findings in some patients with IgAN were advanced and showed chronic lesions, which reflected the difference in renal dysfunction and pathological severity between patients with HSPN and IgAN. Therefore, the timing of the RB and pretreatment with ST may affect the results in the current study. Hilhorst et al. also pointed out that differences in clinical presentation between these 2 diseases were due to the interval between disease onset and the time of RB [43]. However, intriguingly, glomerular endothelial injury with fibrin deposition was apparently predominant in patients with HSPN even though more of these patients received ST before RB. Similarly, recent reports also indicated that endothelial injury and fibrin deposition were distinctive characteristics in patients with HSPN rather than IgAN [14, 44], and such findings are generally considered to represent acute lesions in patients

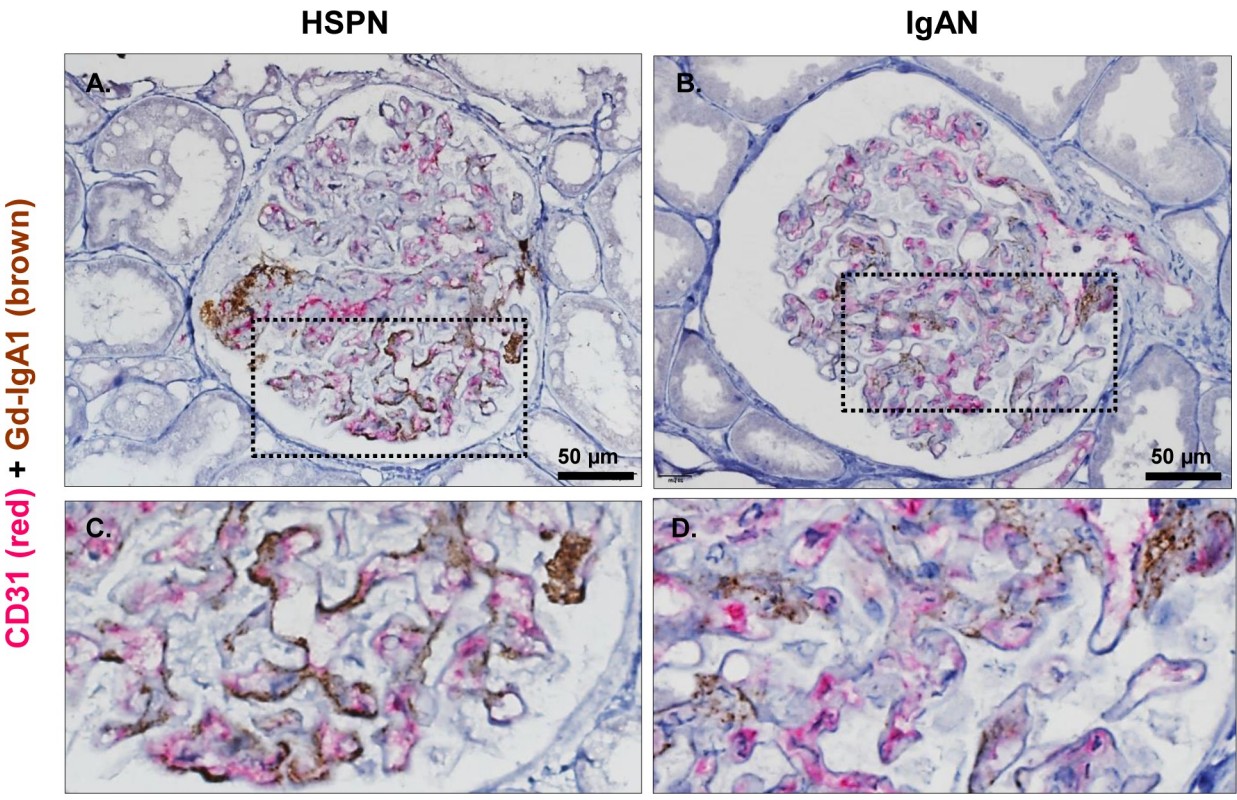

**Fig 5. Double immunostaining for CD31 with Gd-IgA1 between HSPN and IgAN samples.** Paraffin-embedded kidney sections were stained using 2-color immunohistochemistry with CD31 stained red and Gd-IgA1 stained brown. Representative photos of double staining for CD31 with Gd-IgA1 in patients with HSPN (**A** and **C**) or IgAN (**B** and **D**). The boxed area with broken lines in the upper panels (**A** and **B**) is enlarged in the lower panels (**C** and **D**). Original magnification: ×40.

with HSPN [2, 3, 14]. Furthermore, previous research indicated that fibrin itself directly induced inflammation in endothelial cells via its C-terminal end [45]. Wang et al. demonstrated that patients with HSPN with severe fibrin deposition exhibited more severe glomerular damage with active endothelial injury and crescent formation [44]. Taken together, HSPN tends to present as acute glomerular inflammatory lesions with endothelial injury, whereas IgAN tends to develop relative slowly but is associated with progressive mesangial lesions.

With respect to a difference in Gd-IgA1 between the 2 diseases, our findings were consistent with previous reports [25, 26, 28]. Assessing Gd-IgA1 alone was not enough to distinguish between these 2 diseases. However, our study showed several intriguing findings. First, similarly to IgAN, patients with HSPN in histopathologically advanced stage showed high s-Gd-IgA1 levels and evident g-Gd-IgA1 deposition. Second, no correlation between s-Gd-IgA1 and g-Gd-IgA1 was detected in patients with HSPN in spite of the presence of a significant correlation of both forms of Gd-IgA1 in IgAN. Third, in patients with HSPN, significant positive correlations between s-Gd-IgA1 and crescent formation or IL-6 were confirmed, and g-Gd-IgA1 intensity was positively correlated with MCP-1 and IL-8 elevations. In contrast, patients with IgAN showed no correlation of inflammatory cytokines with either type of Gd-IgA1, and both Gd-IgA1 were notably associated with glomerular sclerosis or tubulointerstitial injury. Moreover, serum IgA and IgA/C3 ratios positively correlated with s-Gd-IgA1 in patients with IgAN but not in HSPN. In terms of the influence of ST on both types of Gd-IgA1 in patients with HSPN, no remarkable findings were detected. This lack of findings implies that any potential

bias in our results due to ST for Gd-IgA1 may not be significant. Collectively, both types of Gd-IgA1 in patients with HSPN might serve as biomarkers that reflect disease activity, as shown in the patients with IgAN in our recent report [26]. However, mediators or other factors related to Gd-IgA1 appear to be different between patients with HSPN or IgAN. According to our results, Gd-IgA1 in patients with HSPN might be associated with inflammatory cytokines or acute lesions, whereas that in patients with IgAN might be associated with serum IgA levels or chronic lesions. Additional data are needed to provide reliable evidence of the value of Gd-IgA1 as a biomarker that reflects activity in patients with HSPN.

Another remarkable finding of the present study is that serum inflammatory cytokines were significantly higher in patients with HSPN than in patients with IgAN even though more HSPN patients received ST at the time of RB. In other words, elevation of serum inflammatory cytokines was evident in patients with HSPN, regardless of the anti-inflammatory effects of ST. Thus far, levels for serum inflammatory cytokines were mainly compared among patients with HSP with skin lesions alone and patients with HSP with systemic symptoms [2, 17, 46]. However, to our knowledge, comparisons of serum inflammatory cytokines between patients with HSPN or IgAN are limited. In the present study, serum inflammatory cytokines were higher in patients with HSPN with systemic symptoms, which is consistent with previous reports [2, 17, 46]. Furthermore, elevated serum inflammatory cytokines, especially IL-8 and MCP-1, tended to be associated with intensity of g-Gd-IgA1 deposition and degree of active lesions such as endothelial injury or crescent formation in patients with HSPN. These results led us to presume that elevated levels of serum inflammatory cytokines in HSPN may be related to the formation of Gd-IgA1 and the progression of nephritis. Indeed, a recent review by Heineke et al. mentioned that IL-8 released by vascular endothelial cells promotes an inflammatory response with neutrophil migration and eventually causes systemic small vessel damage [2], which partially supports our findings. However, the contribution of inflammatory cytokines to Gd-IgA1 production in HSPN remains unclear. Basic research to elucidate whether an acute inflammatory response in patients with HSPN affects glycosylation of IgA1 is needed. On the other hand, contrary to our expectations, serum inflammatory cytokines were not elevated in patients with IgAN, and those cytokines levels in IgAN were not associated with both types of Gd-IgA1 or other clinical parameters. However, Suzuki et al. showed that IL-6 in IgA1-producing cell lines derived from the blood of IgAN patients was involved in Gd-IgA1 production [47]. A recent review emphasized the significance of T-cell–derived inflammatory cytokines on the progression of IgAN [48]. Thereby, further studies should evaluate if serum levels of inflammatory cytokines are associated with both types of Gd-IgA1 or disease activity in patients with IgAN.

Next, we discuss localization of the Gd-IgA1 deposition in glomerulus. According to our results, distribution of the g-Gd-IgA1-deposition was dominate in mesangial area in both HSPN and IgAN, which was reasonable when considering a dedicated receptor like mesangial transferrin receptor (CD71) for Gd-IgA1 [17, 46]. However, the patients with HSPN also exhibited considerable glomerular subendothelial IgA deposition, implying that most of the subendothelial IgA-IC in HSPN may be consisted of components other than Gd-IgA1 unlike the mesangial IgA-IC in IgAN. To our knowledge, there have not been similar analysis or discussion, but previous basic researches indicated supportive results to us. First of all, the molecular size of circulating IgA1-IC was reported to be different between patients with HSPN and IgAN [14, 43]. Then, recent review mentioned as follow: IgA1 antibodies in IgAV are generated against autoantigens on endothelial cells, such as infection-related microorganisms or $\beta_2$GPI, and such anti-endothelial IgA1 antibodies (AECA) ultimately induce vessel damage via infiltration of neutrophils [2, 49]. Thus, our hypothesis might be partially plausible, and significant glomerular endothelial IgA in patients with HSPN may also represent AECA, which may

cause endothelial injury with fibrin deposition. However, the meaning of subendothelial Gd-IgA1 deposition as partially seen in our results is unclear, and further analysis focusing on AECA in IgAV or glomerular endothelial Gd-IgA1 deposition in patients with HSPN or IgAN is needed.

Our study had several limitations. First, data from this study were acquired from a single center. Second, the number of enrolled patients was too small to establish strong evidence. Third, the present study mainly targeted patients with less severe HSPN. In fact, none of the patients with HSPN presented with ISKDC grade IV or V. Finally, the influence of ST on our results, especially data for both-Gd-IgA1 and inflammatory cytokines, was not completely revealed in patients with HSPN.

In conclusion, assessing both types of Gd-IgA1 alone was insufficient to distinguish between HSPN and IgAN, but patients with HSPN showed considerable glomerular endothelial injury with subendothelial IgA deposition and significant elevation of serum inflammatory cytokines. Furthermore, such glomerular subendothelial IgA might not necessarily be Gd-IgA1 itself, and factors associated with Gd-IgA1 were inconsistent among these 2 diseases. Taken together, HSPN exhibits not only Gd-IgA1-related glomerulonephritis, similar to IgAN, but also considerable glomerular inflammatory capillaritis distinct from IgAN, and developmental mechanisms for IgAN might not apply to HSPN completely. We suggest that these 2 immuno-pathologically indistinguishable diseases still have different aspects that might hinder to be categorized as variants of one disease.

## Supporting information

**S1 Fig. Correlations between g-Gd-IgA1 intensity and s-Gd-IgA1 level.** Scatter plots of correlations between g-Gd-IgA1 positivity and s-Gd-IgA1 levels in patients with HSPN (**A**) and IgAN (**B**). Data were statistically analyzed using Spearman correlations.
(PDF)

**S2 Fig. Both types of Gd-IgA1 among HSPN patients with or without steroid therapy at the time of renal biopsy.** Comparisons of s-Gd-IgA1 levels (**A**) and g-Gd-IgA1 positivity (**B**) among MCD patients, HSPN patients who received steroid therapy [HSPN-ST (+)], HSPN patients who did not receive steroid therapy [HSPN-ST (-)], and IgAN patients. Horizontal solid lines represent means. Data were statistically analyzed using Kruskal-Wallis tests and Mann-Whitney U tests. $^{*}P<0.05$, $^{**}P<0.01$, and $^{***}P<0.001$. Scatter plots of correlations between g-Gd-IgA1 positivity and s-Gd-IgA1 levels in HSPN-ST (+) (**C**) and HSPN-ST (-) (**D**). Data were statistically analyzed using Spearman correlations.
(PDF)

**S3 Fig. Serum inflammatory cytokines determined by ELISA among HSPN patients with or without steroid therapy at the time of renal biopsy.** Comparison of serum IL-8 (**A**), MCP-1 (**B**), TNF-α (**C**), and IL-6 (**D**) levels among MCD patients, HSPN patients who received steroid therapy [HSPN-ST (+)], HSPN patients who did not receive steroid therapy [HSPN-ST (-)], and IgAN patients. Values are presented as means ± SEM. Data were statistically analyzed using Kruskal-Wallis tests and Mann-Whitney U tests. $^{*}P<0.05$, $^{**}P<0.01$, and $^{***}P<0.001$.
(PDF)

**S4 Fig. Comparisons of both types of Gd-IgA1 among groups based on the Oxford classification of patients with HSPN or IgAN.** Patients with HSPN (**A** and **C**) or IgAN (**B** and **D**) were assigned to groups according to mesangial hypercellularity, endocapillary hypercellularity, segmental glomerulosclerosis, and tubular atrophy/interstitial fibrosis. Values are

presented as means ± SEM. Data were statistically analyzed using Mann-Whitney U tests.
$^*P<0.05$ and $^{**}P<0.01$.
(PDF)

**S5 Fig. Serum inflammatory cytokines determined by ELISA among HSPN patients with or without any systemic symptoms other than nephritis.** Comparison of serum IL-8 (**A**), MCP-1 (**B**), TNF-α (**C**), and IL-6 (**D**) levels between patients with HSPN without any systemic symptoms other than nephritis and patients with HSPN with arthritis or abdominal symptoms (HSPN-AA). Values are presented as means ± SEM. Data were statistically analyzed using Mann-Whitney U tests.
(PDF)

**S6 Fig. Comparisons of serum inflammatory cytokines among the HSPN patients with or without mesangial hypercellularity, segmental glomerulosclerosis, and tubular atrophy/ interstitial based on the Oxford classification.** Comparison of serum IL-8 (**A**, **E** and **I**), MCP-1 (**B**, **F** and **J**), TNF-α (**C**, **G** and **K**), and IL-6 (**D**, **H** and **L**) in patients with HSPN according to the presence of mesangial hypercellularity, segmental glomerulosclerosis, and tubular atrophy/interstitial fibrosis based on the Oxford classification. Values are presented as means ± SEM. Data were statistically analyzed using Mann-Whitney U tests.
(PDF)

**S1 Table. Correlation between both types of Gd-IgA1 and inflammatory cytokines in HSPN patients with or without steroid therapy at the time of renal biopsy.**
(RTF)

**S1 Dataset. Original data.**
(XLSX)

## Acknowledgments

We greatly appreciate the excellent technical assistance provided by Ms. Tomoko Suzuki.

## Author Contributions

**Conceptualization:** Motonori Sugiyama, Yukihiro Wada, Takanori Shibata.

**Data curation:** Motonori Sugiyama, Yukihiro Wada, Nobuhiro Kanazawa, Shohei Tachibana, Taihei Suzuki, Kei Matsumoto.

**Formal analysis:** Motonori Sugiyama, Yukihiro Wada, Nobuhiro Kanazawa, Kei Matsumoto.

**Investigation:** Motonori Sugiyama, Yukihiro Wada, Takanori Shibata.

**Methodology:** Yukihiro Wada, Masayuki Iyoda, Hirokazu Honda, Takanori Shibata.

**Writing – original draft:** Yukihiro Wada.

**Writing – review & editing:** Yukihiro Wada, Takanori Shibata.

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
