## [Decision Letter · Decision Letter 0]

10 Feb 2020

PONE-D-20-00737

A cross-sectional analysis of clinicopathologic similarities and differences between Henoch-Schönlein purpura nephritis and IgA nephropathy

PLOS ONE

Dear Dr Wada,

Thank you for submitting your manuscript to PLOS ONE. After careful consideration, we feel that it has merit but does not fully meet PLOS ONE’s publication criteria as it currently stands. Therefore, we invite you to submit a revised version of the manuscript that addresses the points raised during the review process.

In particular, as highlighted from Reviewer 1, the influence of corticosteroid therapy on the showed data should be addressed and discussed.

We would appreciate receiving your revised manuscript by Mar 26 2020 11:59PM. To enhance the reproducibility of your results, we recommend that if applicable you deposit your laboratory protocols in protocols.io, where a protocol can be assigned its own identifier (DOI) such that it can be cited independently in the future. For instructions see: http://journals.plos.org/plosone/s/submission-guidelines#loc-laboratory-protocols

We look forward to receiving your revised manuscript.

Kind regards,

Fabio Sallustio

Academic Editor

PLOS ONE

Journal Requirements:

Reviewers' comments:

Reviewer's Responses to Questions

**Comments to the Author**

1. Is the manuscript technically sound, and do the data support the conclusions?

Reviewer #1: Partly

Reviewer #2: Yes

2. Has the statistical analysis been performed appropriately and rigorously? 

Reviewer #1: N/A

Reviewer #2: Yes

3. Have the authors made all data underlying the findings in their manuscript fully available?

Reviewer #1: No

Reviewer #2: Yes

4. Is the manuscript presented in an intelligible fashion and written in standard English?

Reviewer #1: No

Reviewer #2: Yes

5. Review Comments to the Author

Reviewer #1: Sugiyama et al have carried out a comparative analysis of the clinicopathologic findings between 24 adult patients with Henock-Schonlein Purpura nephritis (HSPN) and 56 individuals affected by IgA Nephropathy (IgAN).

Serum gal-deficient-IgA1 (s-Gd-IgA1) were measured using KM55 ELISA kit and glomerular deposits of Gd-IgA1 (g-Gd-IgA1) were detected using KM55 antiserum. Moreover, the authors measured some inflammatory cytokines (IL-6, IL-8, TNFα and MCP-1) in the serum samples of all patients.

Results demonstrated similar amounts of Gd-IgA1 deposits in the glomeruli (g) of patients with HSPN and those with IgAN. But g-Gd-IgA1 deposits were more evident in the advanced phase of both diseases. Next, the authors found significant correlations between s-Gd-IgA1 and IL-6 in the serum samples and between g-Gd-IgA1 and IL-8 only in the patients with HSPN. In conclusion, the authors evidenced more inflammation in HSPN (glomerular nephritis) than IgAN patients. They concluded that the meccanisms of development of both diseases are not completely identical.

Results of this paper demonstrate that Gd-IgA1 deposits are present in both diseases but patients with HSPN have considerable glomerular endothelial injury caused by IgA1 deposits and significant increased levels of some inflammatory cytokines.

The paper has some bias that should be evidenced in the discussion

1. The measurement of inflammatory cytokines was carried out in serum samples of HSPN patients under corticosteroid therapy. This means that the serum levels of some cytokines were normal because patients received therapy that can modify the cytokine values.

2. The presence of a few significant correlations between s-Gd-IgA1 and clinicopathologic findings in HSPN may be explained by corticosteroid therapy

3. Correlations between significant of IL-8 and g-Gd-IgA1 are not significant (p=0.052)

4. Fig. 3 shows no statistical difference of serum IL-8 between E0 and E1 (p<0.07)

5. s-Gd-IgA1 and g-Gd-IgA1 may be considered biomarkers of HSPN in active phase only in presence of capillaritis (positivity of CD31).

6. p25 line2 fluctuation of g-Gd-IgA1 deposits and active lesion have not been demonstrated because this is not a longitudinal study.

Minor point

P9 line3 delete and

Reviewer #2: In this paper, authors aimed to clarify the clinic-pathologic differences between Henoch-Schönlein purpura nephritis and IgA nephropathy. They analyzed data from 24 HSPN and 56 IgAN in a period going from 2008 to 2018.

They measured serum levels of IL-8, MCP-1, TNF-α, and IL-6, levels of s-Gd-IgA1 and g-Gd-IgA1-deposition. They tested clinical characteristics and histological parameters, according to ISKDC) classification for HSNP and Oxford classification for IgAN.

Although the absence of relevant findings, the study design was well assessed and the statistical analysis were well performed, giving strength to the result. For these reasons, the data are worthy of publication.

Minor Points:

- In the tables 1, 2, 3 and 4 statistically significant P value should be highlighted by the use of bold

- Page 9 line 4: delete “and”

- I suggest to show the p values in the legend and not in the figures, for a better representation of the data

- Most important concern: in figure 4, I suggest using arrows to help readers to discriminate between mesangial and endothelial area.

6. PLOS authors have the option to publish the peer review history of their article (what does this mean?). If published, this will include your full peer review and any attached files.

Reviewer #1: No

Reviewer #2: No

---

## [Author Response · Author response to Decision Letter 0]

25 Mar 2020

March 25, 2020

Dr. Fabio Sallustio

Academic Editor

PLOS ONE

RE: PONE-D-20-00737

A cross-sectional analysis of clinicopathologic similarities and differences between Henoch-Schönlein purpura nephritis and IgA nephropathy

Dear Dr. Sallustio:

We appreciate the meticulous review of our study and are pleased to learn that our submission is of interest. We appreciate the opportunity to submit a revised manuscript. All changes, including major revisions for the problems listed below and minor modifications for English grammar, are shown in red in the revised manuscript using track changes. Furthermore, the data-set (excel sheet) including raw data and summary of the analysis in the present study was submitted as supporting information to ensure that all data underlying the findings in our manuscript are fully available.

We have addressed the concerns raised as follows:

Journal requirements

1. No changes are required regarding our financial disclosure.

2. To enhance the reproducibility of your results, we recommend that if applicable you deposit your laboratory protocols in protocols.io, where a protocol can be assigned its own identifier (DOI) such that it can be cited independently in the future. 

Because the experimental protocols are clearly detailed in the materials and methods section, we did not prepare a separate specific laboratory protocol.

We carefully read PLOS ONE's style requirements and prepared the revised manuscript and file names based on the instructions.

Reviewer 1

Major comments

Sugiyama et al have carried out a comparative analysis of the clinicopathologic findings between 24 adult patients with Henock-Schonlein Purpura nephritis (HSPN) and 56 individuals affected by IgA Nephropathy (IgAN). Serum gal-deficient-IgA1 (s-Gd-IgA1) were measured using KM55 ELISA kit and glomerular deposits of Gd-IgA1 (g-Gd-IgA1) were detected using KM55 antiserum. Moreover, the authors measured some inflammatory cytokines (IL-6, IL-8, TNFα and MCP-1) in the serum samples of all patients. Results demonstrated similar amounts of Gd-IgA1 deposits in the glomeruli (g) of patients with HSPN and those with IgAN. But g-Gd-IgA1 deposits were more evident in the advanced phase of both diseases. Next, the authors found significant correlations between s-Gd-IgA1 and IL-6 in the serum samples and between g-Gd-IgA1 and IL-8 only in the patients with HSPN. In conclusion, the authors evidenced more inflammation in HSPN (glomerular nephritis) than IgAN patients. They concluded that the meccanisms of development of both diseases are not completely identical. Results of this paper demonstrate that Gd-IgA1 deposits are present in both diseases but patients with HSPN have considerable glomerular endothelial injury caused by IgA1 deposits and significant increased levels of some inflammatory cytokines.

The paper has some bias that should be evidenced in the discussion

1. The measurement of inflammatory cytokines was carried out in serum samples of HSPN patients under corticosteroid therapy. This means that the serum levels of some cytokines were normal because patients received therapy that can modify the cytokine values.

Thank you very much for this important comment. To address this concern, we compared levels of inflammatory cytokines among HSPN patients who received steroid therapy (ST) [HSPN-ST (+), n=9] and HSPN patients who did not receive ST [HSPN-ST (-), n=15] at the point of renal biopsy (RB). As shown in the newly prepared S3 Fig A, serum concentrations (mean ± SEM) of IL-8 were significantly higher in HSPN-ST (-) compared to HSPN-ST (+) (86.5 ± 38.1 vs. 23.4 ± 13.9 pg/mL, respectively; p = 0.034). Serum levels of MCP-1, TNF-α, and IL-6 were not significantly different between groups (S3 Fig B, C, and D) (Page 17, Line 14-19).

At first, we hypothesized that all inflammatory cytokines measured would be significantly higher in HSPN-ST (-) compared to HSPN-ST (+) because of the anti-inflammatory effect of ST. However, except for IL-8, our results did not support this hypothesis. This may be attributable to the small number of patients, diversity of the dose of ST, or difference of administration period of ST in each case.

We added a new S3 Fig as a supporting file in response to your question and mentioned the involvement of ST in the discussion (Page 26, Line 7-9). Furthermore, we described the influence of ST as a limitation of the study (Page 28, Line 9-10).

2. The presence of a few significant correlations between s-Gd-IgA1 and clinicopathologic findings in HSPN may be explained by corticosteroid therapy

As mentioned, we need to address the influence of ST for HSPN on the results of the present study. In particular, we focus on the influence of ST on the findings for both types of Gd-IgA1 in HSPN. In the present study, 9 of 24 patients with HSPN had already received ST against purpura rather than GN at the point of RB (Page 11, Line 23-Page 12, Line 1). No patients with HSPN had completed ST against purpura before undergoing RB.

The newly prepared S2 Fig shows the levels of s-Gd-IgA1 and the intensity of g-Gd-IgA1 deposition between [HSPN-ST (+), n=9] and [HSPN-ST (-), n=15]. As shown in S2 Fig A, s-Gd-IgA1 levels tended to be higher in HSPN-ST (+) compared to HSPN-ST (-), although the differences were not statistically significant. Values of g-Gd-IgA1 positivity were comparable between the groups (S2 Fig B). In addition, similar to the results in S1 Fig A, no correlation between s-Gd-IgA1 levels and g-Gd-IgA1 intensity was detected with either HSPN-ST (+) (S2 Fig C) or HSPN-ST (-) (S2 Fig D). (Page 15, Line 24-Page 16, Line 8). 

In addition, we evaluated the associations between serum inflammatory cytokines and both types of Gd-IgA1 in HSPN-ST (+) and HSPN-ST (-). As shown in the newly prepared S1 Table, serum IL-6 levels showed a significant positive correlation with s-Gd-IgA1 levels in HSPN-ST (+) and a significant positive correlation with g-Gd-IgA1 intensity in HSPN-ST (-). On the other hand, other serum cytokines, including IL-8, MCP-1, and TNF-α, did not correlate with either type of Gd-IgA1 in either group (S1 Table). However, those 3 cytokines showed a tendency to positively correlate with g-Gd-IgA1 intensity in HSPN-ST (-) (Page 20 Line 1-7).

Taken together, additional analysis indicated no remarkable differences between HSPN-ST (-) and HSPN-ST (+). Notable findings were not obtained from HSPN patients in the present study even after we divided these patients according to the presence or absence of ST at the point of RB. Therefore, we consider that the potential bias effect due to ST may not be significant (Page 25, Line 18-21). We described the influence of ST as a limitation of the present study (Page 28, Line 9-10).

3. Correlations between significant of IL-8 and g-Gd-IgA1 are not significant (p=0.052)

Thank you for pointing this out. We did not use bold to highlight the P value in the revised Table 4. Furthermore, we did not use the term “significant” in the explanation regarding correlations between IL-8 levels and g-Gd-IgA1 deposition (Page 3, Line 6-7) (Page 19, Line 17).

4. Fig. 3 shows no statistical difference of serum IL-8 between E0 and E1 (p<0.07)

As you pointed out, Fig 3A did not show significant differences in serum IL-8 levels between E0 and E1 in patients with HSPN (P=0.0667). Thus, we modified Fig 3A and put the P value directly on the figure. In addition, we did not use the term “significant” in the explanation of the results shown in Fig 3A (Page 21, Line 10-11).

5. s-Gd-IgA1 and g-Gd-IgA1 may be considered biomarkers of HSPN in active phase only in presence of capillaritis (positivity of CD31).

As you mentioned, both types of Gd-IgA1 appear to have the potential to be biomarkers that reflect the active phase of HSPN consisting of glomerular capillaritis, crescent formation, and so on. However, it was difficult to strongly suggest this potential from our results because of the limitation of sample size as well as the study design (retrospective, cross-sectional study in a single facility). Furthermore, the absence of capillaritis or crescent depends on the timing of RB and the presence of pretreatment at the time of RB, which could be important confounding factors for reliable evidence. Thus, we described future prospects and tasks needed to establish strong evidence of the value of Gd-IgA1 as a biomarker reflecting active lesions in patients with HSPN (Page 26, Line 3-4)

6. p25 line2 fluctuation of g-Gd-IgA1 deposits and active lesion have not been demonstrated because this is not a longitudinal study.

As you pointed out, this is a retrospective cross-sectional study. Thus, it is hard for this study to explain the fluctuation (time-course) of g-Gd-IgA1 deposits and active lesions in HSPN. We modified the discussion to reflect this point (Page 26, Line 16-17).

Minor point: 

P9 line3 delete and

Thank you so much for your suggestion. We deleted “and” in line 3 of Page 9 (Page 9, Line 6).

Reviewer 2

In this paper, authors aimed to clarify the clinic-pathologic differences between Henoch-Schönlein purpura nephritis and IgA nephropathy. They analyzed data from 24 HSPN and 56 IgAN in a period going from 2008 to 2018. They measured serum levels of IL-8, MCP-1, TNF-α, and IL-6, levels of s-Gd-IgA1 and g-Gd-IgA1-deposition. They tested clinical characteristics and histological parameters, according to ISKDC) classification for HSNP and Oxford classification for IgAN. Although the absence of relevant findings, the study design was well assessed and the statistical analysis were well performed, giving strength to the result. For these reasons, the data are worthy of publication.

Minor Points: 

1. In the tables 1, 2, 3 and 4 statistically significant P value should be highlighted by the use of bold

As you suggested, we highlighted significant P values using bold text in Tables 1, 2, 3, 4, and S1 Table. 

2. Page 9 line 4: delete “and”

We deleted “and” in line 6 of Page 9.

3. I suggest to show the p values in the legend and not in the figures, for a better representation of the data

We edited the explanation of P value in all figures regarding comparisons among study groups (Fig 1, Fig 2, Fig 3, Fig 4, S2 Fig, S3 Fig, and S4 Fig) and included the P values in the figure legends. Descriptions of P values were left in figures showing correlations (S1 Fig and S2 Fig) with sample numbers (n) and R values to help readers easily confirm the findings.

4. Most important concern: in figure 4, I suggest using arrows to help readers to discriminate between mesangial and endothelial area. 

In the revised Fig 4, representative positive findings in the endothelial area are shown by white arrows, and those in mesangial area are shown by white asterisks.

We feel that we have addressed all of the concerns of the referees and that the manuscript has been significantly improved. We hope that you find the revised version of our manuscript suitable for publication in PLOS ONE.

We look forward to receiving your response.

Sincerely,

Yukihiro Wada

Division of Nephrology, Department of Medicine

Showa University School of Medicine

Tokyo, Japan

Address: 1-5-8 Hatanodai, Shinagawa-ku, Tokyo 142-8666, Japan

Tel: +81-3-3784-8533

E-mail address: yukihiro@med.showa-u.ac.jp

---

## [Decision Letter · Decision Letter 1]

3 Apr 2020

PONE-D-20-00737R1

A cross-sectional analysis of clinicopathologic similarities and differences between Henoch-Schönlein purpura nephritis and IgA nephropathy

PLOS ONE

Dear Dr Wada,

Thank you for submitting your manuscript to PLOS ONE. After careful consideration, we feel that it has merit but does not fully meet PLOS ONE’s publication criteria as it currently stands. Therefore, we invite you to submit a revised version of the manuscript that addresses the points raised during the review process.

ACADEMIC EDITOR: 

Just correct some typos

We would appreciate receiving your revised manuscript by May 18 2020 11:59PM. To enhance the reproducibility of your results, we recommend that if applicable you deposit your laboratory protocols in protocols.io, where a protocol can be assigned its own identifier (DOI) such that it can be cited independently in the future. For instructions see: http://journals.plos.org/plosone/s/submission-guidelines#loc-laboratory-protocols

We look forward to receiving your revised manuscript.

Kind regards,

Fabio Sallustio

Academic Editor

PLOS ONE

Reviewers' comments:

Reviewer's Responses to Questions

**Comments to the Author**

1. If the authors have adequately addressed your comments raised in a previous round of review and you feel that this manuscript is now acceptable for publication, you may indicate that here to bypass the “Comments to the Author” section, enter your conflict of interest statement in the “Confidential to Editor” section, and submit your "Accept" recommendation.

Reviewer #1: All comments have been addressed

Reviewer #2: All comments have been addressed

2. Is the manuscript technically sound, and do the data support the conclusions?

Reviewer #1: Yes

Reviewer #2: Yes

3. Has the statistical analysis been performed appropriately and rigorously? 

Reviewer #1: Yes

Reviewer #2: Yes

4. Have the authors made all data underlying the findings in their manuscript fully available?

Reviewer #1: Yes

Reviewer #2: Yes

5. Is the manuscript presented in an intelligible fashion and written in standard English?

Reviewer #1: (No Response)

Reviewer #2: Yes

6. Review Comments to the Author

Reviewer #1: Pag 26 line 16 delete were

Pag 26 line 18 these results let us to presume

After these corrections the revised paper will be suitable for publication.

Reviewer #2: Recommendation: Accept With No Changes

The authors have completely addressed all the issues raised by reviewers

7. PLOS authors have the option to publish the peer review history of their article (what does this mean?). If published, this will include your full peer review and any attached files.

Reviewer #1: No

Reviewer #2: No

---

## [Author Response · Author response to Decision Letter 1]

5 Apr 2020

April 5, 2020

Dr. Fabio Sallustio

Academic Editor

PLOS ONE

RE: PONE-D-20-00737R1

A cross-sectional analysis of clinicopathologic similarities and differences between Henoch-Schönlein purpura nephritis and IgA nephropathy

Dear Dr. Sallustio:

We really appreciate the meticulous review and are pleased to learn that our submission is of interest. We appreciate the opportunity to submit a revised manuscript. All changes, including minor modifications, are shown in red in the revised manuscript using track changes. 

We have addressed the concerns raised as follows:

Journal requirements

1. No changes are required regarding our financial disclosure.

2. To enhance the reproducibility of your results, we recommend that if applicable you deposit your laboratory protocols in protocols.io, where a protocol can be assigned its own identifier (DOI) such that it can be cited independently in the future. 

Because the experimental protocols are clearly detailed in the materials and methods section, we did not prepare a separate specific laboratory protocol.

Reviewer 1

After these corrections the revised paper will be suitable for publication.

1. Page 26 line 16 delete were

Thank you so much for your suggestion. We deleted “were” in line 16 of Page 26.

2. Pag 26 line 18 these results let us to presume

As you pointed out, that sentence appeared to be inappropriate. We modified as follow; These results led us to presume that elevated levels of serum inflammatory cytokines in HSPN may be related to the formation of Gd-IgA1 and the progression of nephritis.

Reviewer 2

Recommendation: Accept With No Changes

The authors have completely addressed all the issues raised by reviewers

We really appreciate your careful review of our manuscript. We completed to modify some inappropriate parts of the manuscript. 

We feel that we have addressed all of the concerns of the referees and that the manuscript has been significantly improved. We hope that you find the revised version of our manuscript suitable for publication in PLOS ONE.

We look forward to receiving your response.

Sincerely,

Yukihiro Wada

Division of Nephrology, Department of Medicine

Showa University School of Medicine

Tokyo, Japan

Address: 1-5-8 Hatanodai, Shinagawa-ku, Tokyo 142-8666, Japan

Tel: +81-3-3784-8533

E-mail address: yukihiro@med.showa-u.ac.jp

---

## [Editor Report · Decision Letter 2]

9 Apr 2020

A cross-sectional analysis of clinicopathologic similarities and differences between Henoch-Schönlein purpura nephritis and IgA nephropathy

PONE-D-20-00737R2

Dear Dr. Wada,

We are pleased to inform you that your manuscript has been judged scientifically suitable for publication and will be formally accepted for publication once it complies with all outstanding technical requirements.

With kind regards,

Fabio Sallustio

Academic Editor

PLOS ONE
---

## [Editor Report · Acceptance letter]

14 Apr 2020

PONE-D-20-00737R2 

A cross-sectional analysis of clinicopathologic similarities and differences between Henoch-Schönlein purpura nephritis and IgA nephropathy 

Dear Dr. Wada:

I am pleased to inform you that your manuscript has been deemed suitable for publication in PLOS ONE. Congratulations! Your manuscript is now with our production department. 

With kind regards,

on behalf of

Dr. Fabio Sallustio 

Academic Editor

PLOS ONE